# Auto-encoding NMR chemical shifts from their native vector space to a residue-level biophysical index

Gabriele Orlando[1,2], Daniele Raimondi[3] & Wim F. Vranken[1,2,4]

Chemical shifts (CS) are determined from NMR experiments and represent the resonance frequency of the spin of atoms in a magnetic field. They contain a mixture of information, encompassing the in-solution conformations a protein adopts, as well as the movements it performs. Due to their intrinsically multi-faceted nature, CS are difficult to interpret and visualize. Classical approaches for the analysis of CS aim to extract specific protein-related properties, thus discarding a large amount of information that cannot be directly linked to structural features of the protein. Here we propose an autoencoder-based method, called ShiftCrypt, that provides a way to analyze, compare and interpret CS in their native, multi-dimensional space. We show that ShiftCrypt conserves information about the most common structural features. In addition, it can be used to identify hidden similarities between diverse proteins and peptides, and differences between the same protein in two different binding states.

[1] Interuniversity Institute of Bioinformatics in Brussels, ULB-VUB, 1050 Brussels, Belgium. [2] Structural Biology Brussels, Vrije Universiteit Brussel, 1050 Brussels, Belgium. [3] ESAT-STADIUS, KU Leuven, 3001 Leuven, Belgium. [4] Center for Structural Biology, VIB, 1050 Brussels, Belgium. Correspondence and requests for materials should be addressed to W.F.V. (email: Wim.Vranken@vub.be)

Chemical shifts (CS) are the primary data gained from nuclear magnetic resonance (NMR) experiments. They reflect the resonance frequency of the spin of atoms in a magnetic field, and relate the peaks in NMR spectra to specific atoms within molecule(s) under study. The CS value of an atom is determined by the actual magnetic field experienced by that atom, and is influenced by its local environment, both in terms of its chemical bonding and of other atoms surrounding it in space. The CS values observed for atoms in a molecule can therefore give insights into the conformations that the molecule adopts in solution.

In structural biology, many proteins have been studied by NMR over the last few decades. Because CS values are of key importance when studying proteins at atomic resolution[1], they are relatively abundantly available, especially because of the efforts of the BioMagResBank in storing and curating protein NMR data[2]. Many studies have demonstrated the relationship between the CS values of the atoms in an amino acid residue and the conformational characteristics of that amino acid in a given protein: these CS values are related to the angles the backbone of the protein can adopt in its folded state[3], to the backbone flexibility[4], to solvent accessibility[5], and they can be used to assign secondary structures[6] or secondary structure propensities[7]. Moreover, since NMR experiments can be performed on proteins in solution, the related data are extremely relevant for the study of intrinsically disordered and non-globular polypeptides[8].

Due to the intrinsic multidimensional nature of chemical shift values, major challenges remain in their practical interpretation. For example, the CS values of two amino acids of a different type that experience the same local environment will differ, because the chemical composition of the amino acids is not the same. Alternatively, the same chemical shift values could be observed for atoms in two distinct amino acids of the same type, even if these amino acids experience a very different local environment. Currently, CS values are typically used to directly estimate specific biophysical properties. However, this approach only allows the interpretation of a single characteristic per time, and will so neglect important cross-correlated features. This could be avoided by employing a "hands-off" approach to the CS values, through direct interpretation of their values, without explicitly associating them to biophysical characteristics.

Chemical shift information can be interpreted through two-dimensional (2D) correlations, considering the relation between the CS values of two atoms at the same time. While this approach is simple and easy to interpret, data cannot always be represented in 2D or even 3D space. Higher-dimensional space is however nearly impossible to understand for humans, as our brain evolved to interact with 3D environments. On the other side, data are often naturally multidimensional, and although their visualization might pose a great challenge, it is relatively easy to analyze them by using the proper mathematical framework. Neural networks, for example, are mathematical entities that can operate on vector spaces with an arbitrary number of dimensions; they are therefore capable of analyzing and interpreting higher-dimensional data in their native space, without requiring the translation into an artificially constructed space with reduced dimensionality, which inevitably causes loss of information and structure in the data. Once the information has been formalized into a mathematical representation, such as points in a multidimensional space, neural networks can extract information from the data and perform complex elaborations of such a meaningful dimensionality reduction with minimal information loss, mapping the original data to a lower-dimensional space, helping its visualization and understanding.

Here, we propose a method to encode the chemical shift information of the atoms per amino acid residue into a single abstract value termed the ShiftCrypt index. This index is computed with a three-layer autoencoder neural network, which is structured as two mirrored sub-modules. One, the encoder, encrypts the input CS in a single value (the ShiftCrypt index), while the second one, the decoder, tries to reconstruct the original CS from the aforementioned single value. The neural network optimizer minimizes the differences between the decoded and the original CS.

We show that the single value encoded per amino acid is highly correlated with the biophysical structural and dynamic features of that amino acid, even if the encoding itself did not include any such information. The ShiftCrypt index is largely residue independent, and has the characteristics of a "hidden" biophysical feature, summarizing the chemical shift data and translating it into a mono-dimensional attribute. Since it summarizes the CS information content, it can be used to directly compare the similarities and differences in the chemical shift information between diverse proteins and peptides, or in the same protein observed in different states.

## Results

**ShiftCrypt as a biophysical feature of in-solution proteins**. With ShiftCrypt, we approach the data from a different point of view compared with classical approaches. Instead of looking at the data from the familiar protein conformation perspective, the ShiftCrypt index focuses on the native CS data, using the multidimensional CS as reference space. In other words, we interpret the CS values from NMR experiments as referring to a non-structural, multidimensional space that is connected to latent biophysical information, instead of translating the CS data in reference to three-dimensional atom coordinate space. ShiftCrypt performs a pure mathematical transformation that provides a single per-residue value that serves as a latent feature of in-solution proteins. This feature gives information about the overall amino acid behavior in a single dimension, based on the intrinsically multidimensional chemical shift information. The ShiftCrypt value is not influenced by, or based on, any derived (i.e., structural) data, and is a purely experimental index. In the following sections, we show the major properties of this latent biophysical feature and its relationship with classical structural properties of proteins. We also show how the use of this index can allow an easier identification of the structural and functional difference between proteins or different states of the same protein.

**Secondary structure elements**. The connection between chemical shift information and the secondary structure of an amino acid in a protein has been long recognized and exploited[7]. The regular and constrained conformation of the protein backbone in a secondary structure element results in the atoms of the amino acids experiencing similar local environments, which fundamentally differ between the distinct types of secondary structure. This property is well preserved in the ShiftCrypt index, as shown in Supplementary Fig. 1, which shows the relationship of the ShiftCrypt with a secondary structure, as observed for all amino acids in 3385 structures. Supplementary Fig. 2 shows that the behavior of the different amino acid types is similar. To quantify the similarity, we calculated the integral of the superimposition of the distribution of the secondary structure elements with respect to the ShiftCrypt index (Supplementary Table 1). This indicates how well the ShiftCrypt index distinguishes between secondary structure types at the individual residue level.

Supplementary Fig. 2 also indicates that different amino acids have slightly divergent distributions of secondary structure probabilities with respect to the ShiftCrypt index. For example,

the coil distribution for threonines is much tighter than for lysines. This is not due to the different secondary structure propensity of the residue, but is instead related to the amount of information needed to encode the coil CS information. In other words, the larger the portion of the ShiftCrypt index occupied by a secondary structure, the more heterogeneous the CS are within that secondary structure type.

Another interesting characteristic of this index is its low dependence on any particular atom type. Typically, $C^\alpha$ and $C^\beta$ CS are the most indicative atom types[9], but as shown in Supplementary Tables 2, 3, and 4, removing $C^\alpha$, $C^\beta$, or both $C^\alpha$ and $C^\beta$ from the atoms used to calculate the ShiftCrypt index does not strongly affect the separation between the secondary structures based on the ShiftCrypt index.

**Torsion angles**. Given their relation to secondary structure elements and the local environment, the CS of backbone atoms of amino acids are further specifically used to estimate the backbone $\Phi$ and $\Psi$ torsion angles of amino acids in a given protein[3]. This is another feature that is well conserved in the ShiftCrypt index. Figure 1 shows the distribution of the $\Phi$ and $\Psi$ angles in relation to the relative ShiftCrypt index.

Supplementary Figs. 3, 4, 5, and 6 also show that the behavior of the four classes of amino acid (basic, acid, polar, and non-polar) is very similar. While the relationship between the ShiftCrypt index and angles is evident, this effect could simply derive from the fact that the ShiftCrypt index discriminates only the secondary structure elements. Figure 2 shows how, per individual secondary structure type as observed in the related fold, amino acids associated with high and low values of the ShiftCrypt index within that secondary structure class still behave diversely, with a distinction between the alpha-helical and $3_{10}$ helix regions, the turn/helix and extended coil regions, and the antiparallel beta-sheet regions.

**ShiftCrypt correlates with early-folding events**. The Start2Fold database contains information about which residues in proteins are the first to start folding, when the proteins folds from a statistical chain state, as studied by hydrogen/deuterium exchange (HDX) experiments[10]. These residues are more central to the final fold, and are more conserved during evolution, with their behavior more likely to be determined by local interactions with amino acids close in the protein sequence[11]. We identified chemical shift data for more than 60% of the entries in this database, and calculated the ShiftCrypt index for a total of 90 early-folding residues (15 in helix, 15 in coil, and 60 in sheet conformation) and 488 non-early-folding ones (149 in helix, 211 in coil, and 128 in sheet conformation). Supplementary Fig. 7 shows the distribution of the ShiftCrypt index value for these two classes, stratified by secondary structure. We tested the significance of the two distributions with a Wilcoxon signed-rank test. The difference is statistically significant only for the early-folding residues in beta-sheet conformation, with a $p$-value of $6.08 \times 10^{-7}$, indicating that the early-folding characteristics of residues that fold into a beta-strand tend to be transferred to the final fold, unlike those in alpha-helices and coil.

**ShiftCrypt correlates with solvent accessibility**. The solvent accessibility of the residues of a protein can be estimated from CS[5], with values tending toward typical "random coil" ones for solvent-exposed residues[12]. This property of the folded structure

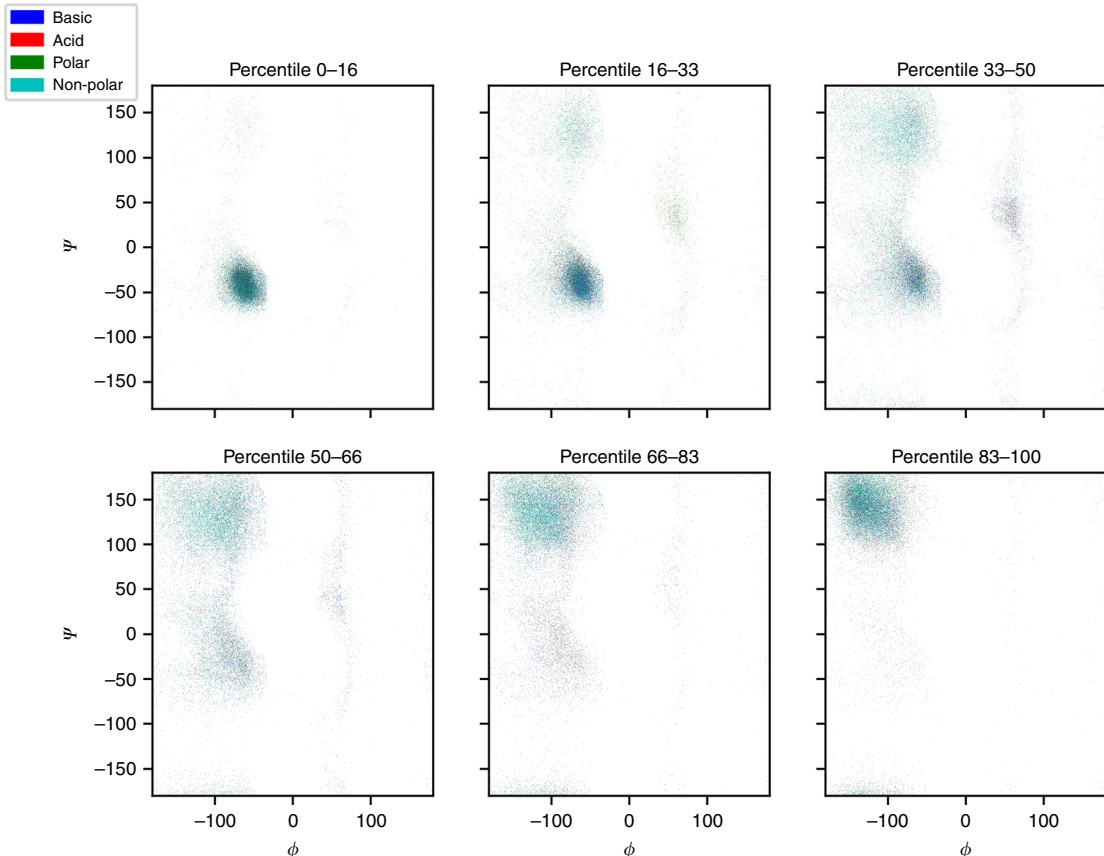

**Fig. 1** Relation between the torsion angles and the Shiftcrypt index. Every plot represents an interval of the ShiftCrypt index. The amino acid types are defined by color: blue for basic, red for acid, green for polar, and cyan for non-polar

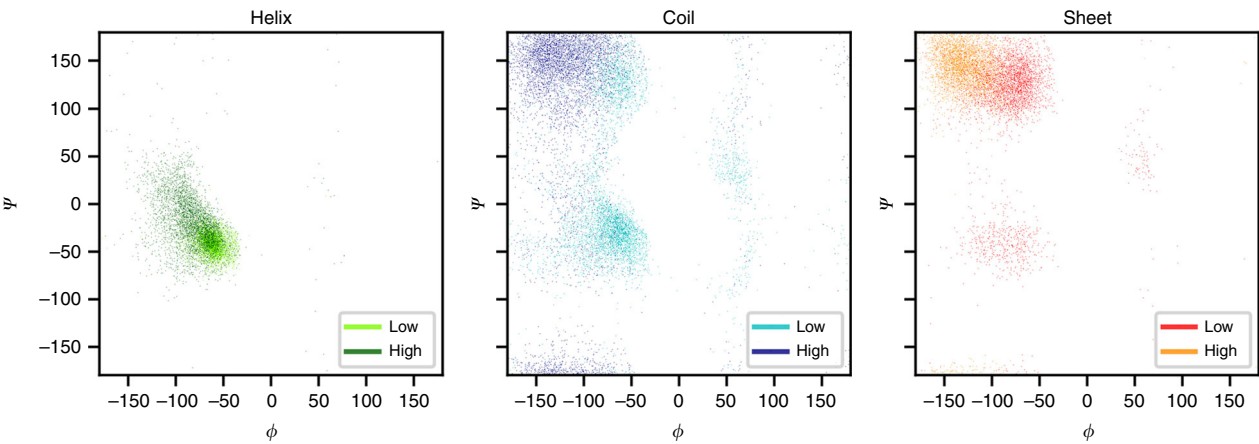

**Fig. 2** Torsion angles distribution for extreme ShiftCrypt values. Torsion angles distribution for low (dark) and high (light) values of ShiftCrypt, stratified by secondary structure. Only the residues for which the same secondary structure is assigned to the residue in question in all the NMR models are taken into consideration

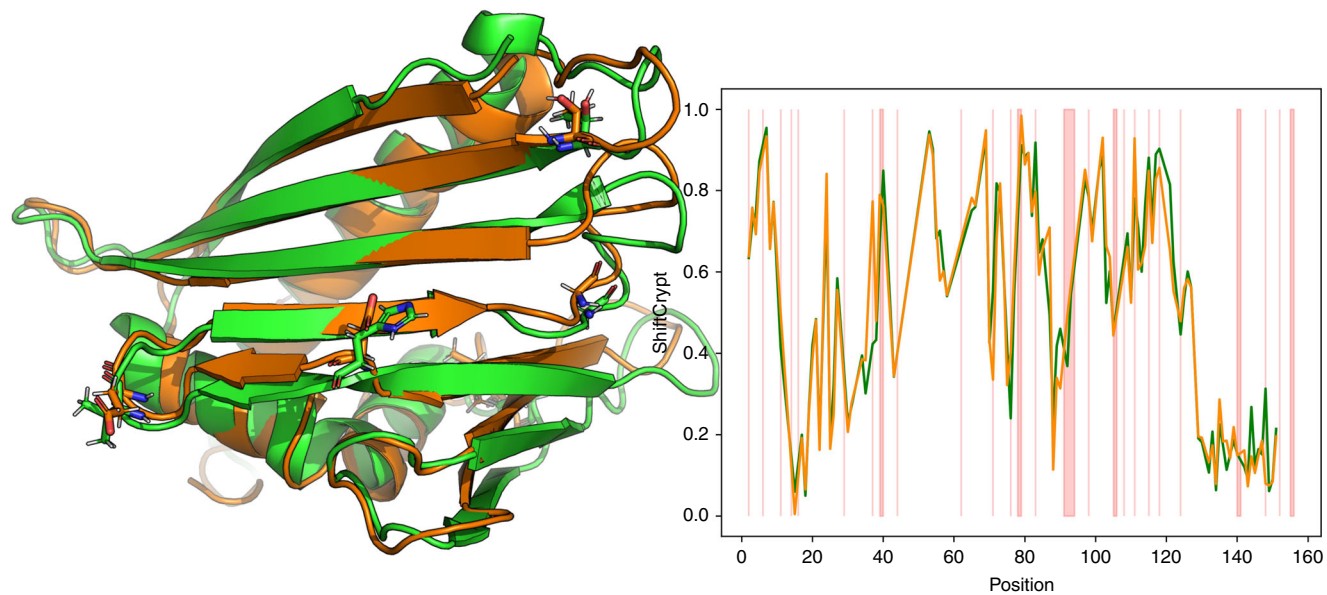

**Fig. 3** Comparison of the ShiftCrypt index for two homologous proteins. The left side of the figure shows the superimposition of the cherry (orange) and strawberry (green) allergen structures. On the right side, we show their respective ShiftCrypt profiles, with the red bars highlighting amino acid mismatches in the alignment of the two proteins

is also reflected by the ShiftCrypt index, as shown in Supplementary Fig. 8, which shows the distribution for buried and exposed amino acids, stratified by secondary structure. The $p$-values obtained with a Wilcoxon signed-rank test for the alpha-helix, coil, and beta-sheet secondary structures are $1.11 \times 10^{-64}$, $1.08 \times 10^{-9}$, and $1.12 \times 10^{-232}$, respectively. Residues in the core of the protein, which are typically very rigid, tend to have ShiftCrypt index values closer to 1 or 0. Residues that are more exposed, and expected to be more dynamic, have values closer to 0.5. For helices, this difference is the least pronounced, whereas for coil residues, the sharp distribution around 0.5 reflects previous studies[12].

**An illustration of ShiftCrypt in comparing proteins**. Two distinct proteins are often compared on the basis of their three-dimensional structure, if available, and in essence, their static structural elements are related to each other. However, especially when comparing more dynamic proteins, it is important to get

insights about the differences and similarities of their in-solution properties, since this can provide important information about differences in their behavior. CS can provide such insights, but they are not straightforward to use when comparing different amino acid types to each other, as for example, for a mutated residue. The only current solution is to use "secondary shifts" for amino acids, where residue-dependent reference chemical shift values are subtracted from the observed chemical shift. This only enables comparison between the same atom type, and introduces the challenge of which reference chemical shift values are the best to use. The ShiftCrypt index circumvents this issue and can be used to easily and directly compare proteins with different amino acid sequences. Figure 3 shows the structure superimposition and the ShiftCrypt profile of two very similar proteins: the strawberry (PDB ID 2LPX) and cherry (PDB ID 1E09) allergens. None of these proteins are present in the training set of ShiftCrypt. These two proteins share a sequence identity of 79%, with similar structures, as indicated by a backbone atom RMSD of 1.756 Å. Despite this similarity, there are the same notable differences in

their structures. Figure 3 shows a portion of the superimposition of the two structures, in relation to the alignment of their Shift-Crypt index. The shaded red areas highlight sequence differences between the two proteins. While for most of the residues the ShiftCrypt index is almost the same, also in sequence positions where the amino acids are different, there are deviations in the index for residues where the local sequence is exactly the same. In order to explore the reasons behind these differences, we analyzed the structure of the positions in which the differences between the ShiftCrypt values of the two homologous proteins significantly diverged from the observed distribution. These positions have been identified, selecting the residues, yielding ShiftCrypt differences with a probability <0.05, assuming a Gaussian distribution. The selected positions are 38, 72, 77, 88, 122, and 149, of which the side-chain atoms are shown in Fig. 3. Interestingly, these residues are located in regions that are likely to behave slightly different in solution: positions 38, 77, and 88 are located in loops, where the dynamics likely play a crucial role, while positions 122 and 149 are located at the very end of a beta-strand and alpha-helix, respectively. Finally, position 122 is reported to be a loop gap in the strand of the cherry allergen structure, suggesting that the behavior of the region might be different in the two homologous proteins. Note that these positions do not necessarily correspond to ill-defined parts of the NMR structure ensembles (Supplementary Fig. 9).

The profiles for these two proteins are therefore not different in regions with a similar structure, even if the amino acid itself is different. On the other hand, the presence of a region with a conserved sequence but with different structural properties makes the index diverge. We highlight that this example is meant to show that ShiftCrypt is conserved in structurally and dynamically similar proteins, and that it can highlight regions that likely behave differently in solution. If the purpose is only to find conserved secondary structure element approaches based on secondary structure propensities, methods such as d2D[7], are likely more suitable, although they lack the finer per-residue detail (i.e., sharp changes between residues) that is present in the ShiftCrypt profiles. As a comparison point, we report the prediction of d2D and SSP[13] for the same proteins in Supplementary Figs. 10 and 11.

**Comparing very diverse proteins**. To further pursue this concept, we investigated what happens to the ShiftCrypt index when the sequence identity between proteins drops below 50%. In this case, while the overall structure is likely conserved, variability will be expected in the biophysical characteristics, such as dynamics, to enable ligand specificity and other organism-specific tasks. As shown in previous work[4], CS are sensitive to such behavioral changes. Figure 4 shows the comparison between the *Saccharomyces cervisiae* (PDB 3F3Q) and *Arabidopsis thaliana* (PDB 1XFL) oxidized thioredoxin 1. As in the previous section, none of these proteins are present in the training set of ShiftCrypt. The sequence identity is 46%, and the structure is well conserved, but structure is only one of the properties that a protein needs to perform its tasks. Figure 4 shows the secondary chemical shift differences for the $C^\alpha$, $C^\beta$, and $H^\alpha$ atoms, where a per-amino acid-type reference value is subtracted from the actual value to try and account for absolute differences in chemical shift value, and the raw chemical shift values for the N and H atoms. Compared with these values, which show differences throughout that are difficult to interpret, the ShiftCrypt values are easily visualized and show high conservation between these proteins, with a similar overall profile, including fine structure. Some regions, such as the N-terminal residues, are very similar despite having different CS values, whereas other regions, notably 4–8, 15–18, 30–32, 43–44, 48–50, and 78–79, show distinct differences. The side chains of

these residues are shown in the structure in Fig. 4. The differences are mainly at the end of structured alpha-helix and beta-sheet segments, indicating that the source of these differences may be caused by different dynamics or conformational populations at these sites. In particular, the difference around position 30 corresponds to a loop and the start of a beta-sheet (on one side) and of an alpha-helix (on the other). The beta-sheet is shorter in the yeast protein, suggesting that the two protein regions may have different in-solution behavior. The differences in the very first part of the protein (position 4–8), correspond to residues adjacent to the active site, suggesting organism-specific causes. Again, the regions indicated by the ShiftCrypt index do not necessarily correspond to ill-defined regions in the NMR structure ensemble (Supplementary Fig. 12).

We also tested the conservation of the Random coil index (RCI), a chemical shift-based estimation of backbone dynamics (Supplementary Fig. 13). The RCI values in general differ between the two proteins, with changes comparable with what is observed for the scaled chemical shift values. Supplementary Figs. 14 and 15 report the secondary structure propensities calculated with the d2D[7] and SSP[13] index, respectively.

**ShiftCrypt index in dimerization**. Since proteins are dynamic entities, they can behave differently, depending on the function they need to perform. In protein complexes, for example, the individual components are expected to behave differently with respect to their monomeric state. This difference affects the chemical shift information, changing the ShiftCrypt index. Figure 5 shows the comparison between the ShiftCrypt profile of the free and bound state of the RimM protein (PDB 3A1P) and its relation to the dimer structure.

The five blue-highlighted residues represent the ones that show the greatest difference in the ShiftCrypt value. As expected, these residues define the interaction patch between the two proteins. In order to provide a more quantitative evaluation of the usefulness of ShiftCrypt in this regard, we performed a larger analysis based on 280 residues taken from four protein complexes, for which the CS for the free and bound state are available (bmrb IDs 16065 and 16066, 5228 and 15397, 10140 and 10139, and 6806 and 6804). We calculated, for the free and bound pairs, the three secondary structure propensities calculated with d2D[7], SSP[13], ShiftCrypt, and a function of the N and H CS, specifically designed for the detection of interaction patches, as described in ref. [14]. The ShiftCrypt version we used for this purpose only takes N, H, and CO chemical shift values as input. We then calculated the per-residue difference of the various indices for the free and bound form. Intuitively, if the index can identify the residues in the interaction patches, the differences will be higher for the amino acids in the patch with respect to the others. We performed a Wilcoxon rank-sum test for all the seven indices in order to test which ones were associated with higher differences in the interaction patches. The only significant results were ShiftCrypt and the N–H-based method ($p$-value $\times 10^{-7}$ for both). This indicates that, while there is no change in the secondary structure propensities, the CS themselves contain information about dimerization, because of changes in their environment and thus in their biophysical behavior. However, the Pearson's correlation coefficient between the differences highlighted by these two methods is only 0.58, and decreases to 0.38 if we only consider the residues in the interaction patches. Supplementary Fig. 16 shows the scatter plot of the values provided by the two methods. This suggests that the two approaches provide two complementary sources of information. To verify this, we built a naive consensus of the two indices, simply summing the scaled N–H values and the ShiftCrypt index.

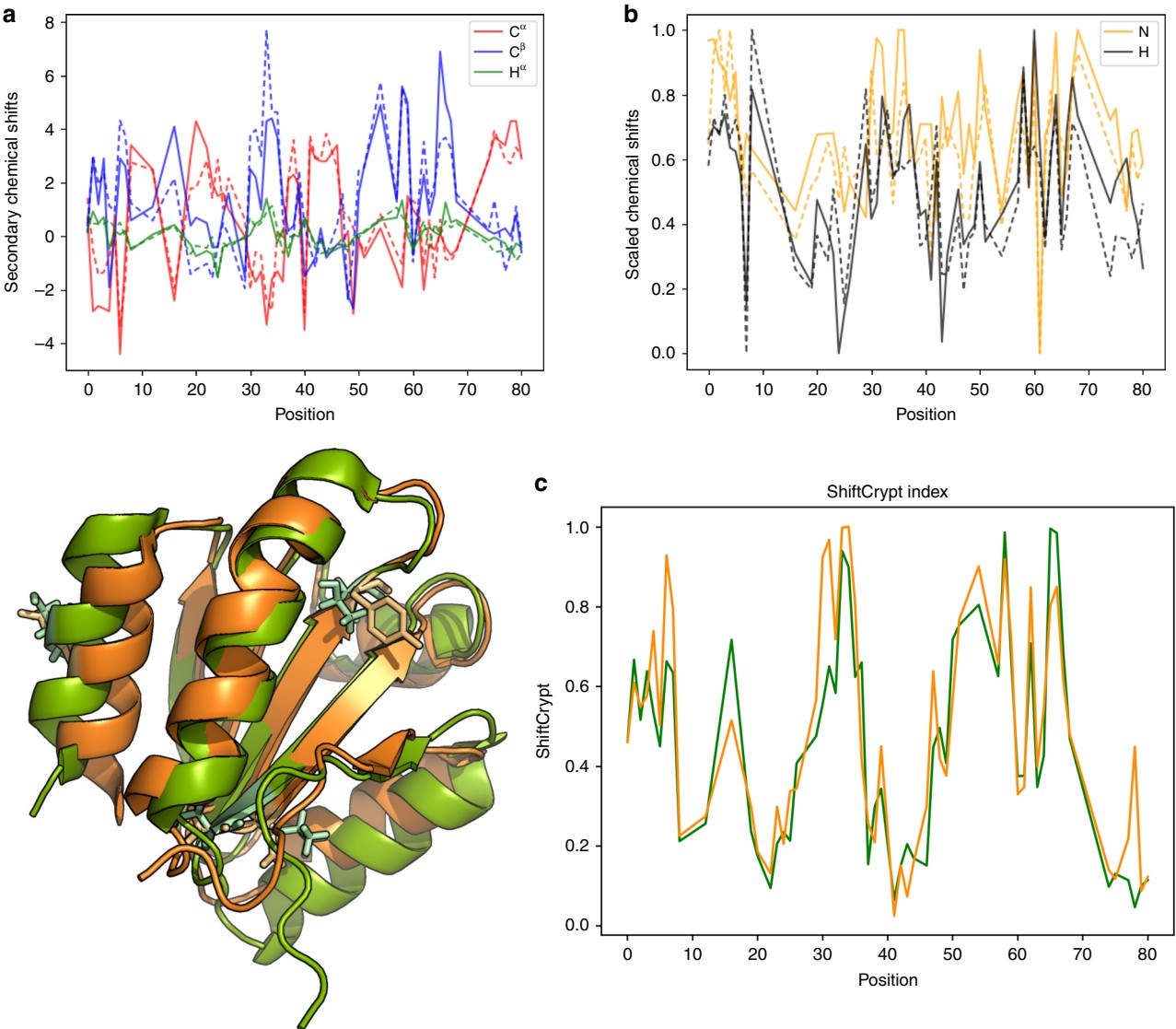

**Fig. 4** Detection of a remote homology using chemical shifts. Panel **a** shows the C$^\alpha$ (red), C$^\beta$ (blue), and H$^\alpha$ (green) secondary chemical shifts, while panel **b** the N-scaled (yellow) and H-scaled (black) scaled chemical shifts for the oxidized thioredoxin 1 of *A. thaliana* (full line) and *S. cerevisiae* (dotted line), which share 46% sequence identity. The ShiftCrypt index (**c**) and superimposed structures are also shown for *A. thaliana* (green) and *S. cerevisiae* (orange)

The resulting *p*-value of this mixed score is reduced to ×10$^{-10}$. This shows that the two methods are complementary and that a more complex machine-learning consensus method may provide even better ways to estimate the patches of interaction. None of the proteins used in this analysis are present in the training set of ShiftCrypt.

**ShiftCrypt index and random coil peptides.** Given the ability of ShiftCrypt to detect the biophysical similarities between amino acids of different types, we also investigated how it assesses the chemical shift values for typical random coil peptides (Fig. 6)[15, 16]. Such peptides are used to describe the default state of amino acids, and their CS are used as the reference to calculate secondary shifts, which are intended to be residue independent. There are, however, indications from molecular dynamics that different amino acid types can have very different default conformational states in such peptides, when referenced in absolute Ramachandran plot terms[17]. The ShiftCrypt index confirms this, with large differences present between the different amino acid types. The residue following the central X amino acid (A, G, and P) only seems to have a major biophysical influence in

very specific cases (e.g., P after V), while the QQXQQ peptide series tends to have higher ShiftCrypt values. The ShiftCrypt value is negatively correlated with the right-handed alpha-helix population (from ref. [17]) and positively with the beta-strand population. This is in line with the protein torsion angle level data, but shows that the ShiftCrypt index captures conformational population information. Overall, these results indicate that the default conformational states of amino acids have a significant biophysical bias, as do by extension the secondary shifts and methods that use them, such as most of the secondary structure propensity predictors.

**Discussion**
ShiftCrypt allows users to visualize the chemical shift data for an amino acid residue by encoding them into a single value. The ShiftCrypt index, through its direct interpretation of the raw chemical shift data, bypasses the often inaccurate mapping of CS on classical structural features. It is therefore, in our view, a more straightforward way to analyze and compare residue-based CS information, especially because it eliminates the absolute differences in CS values between different amino acid types while

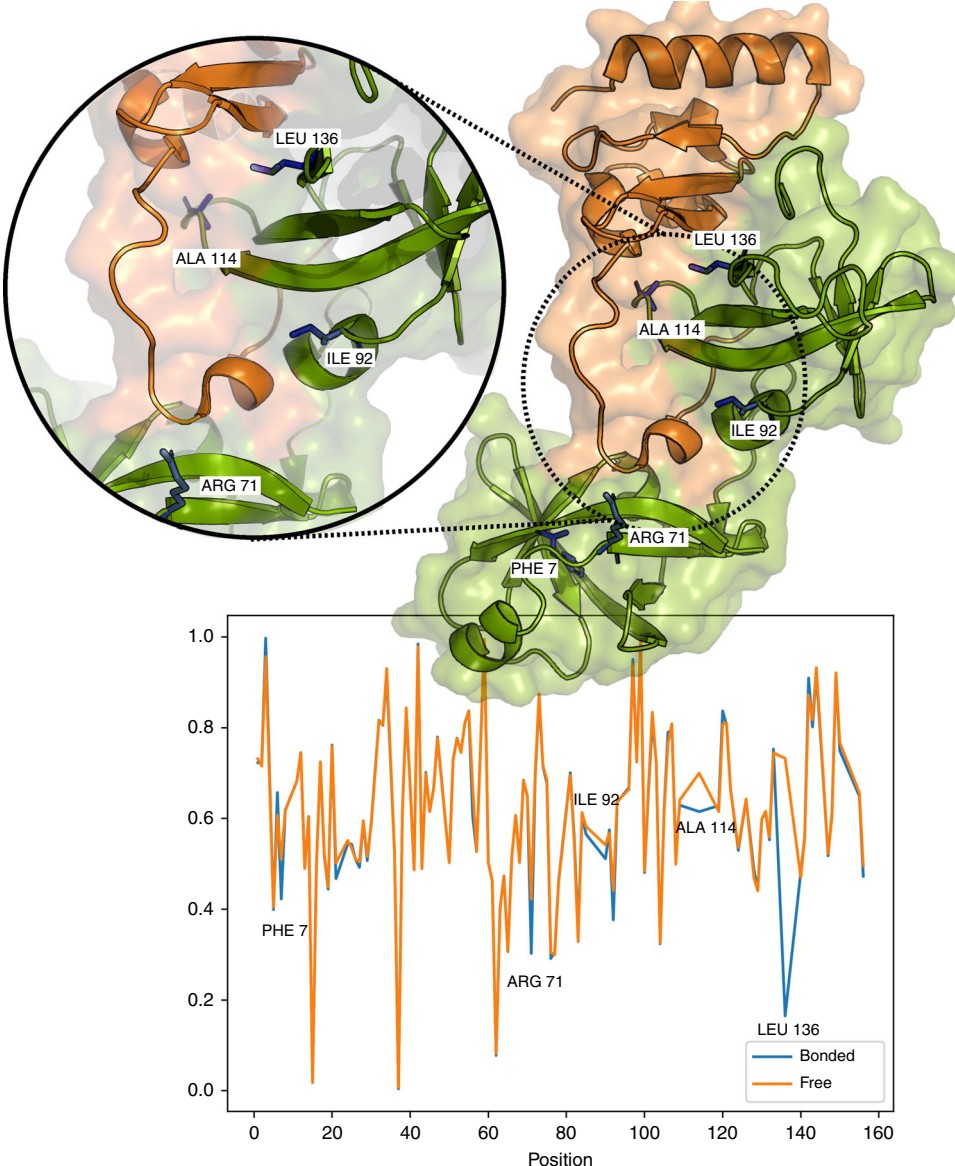

**Fig. 5** Comparison between the free and bonded state of the RimM protein. The bottom part of the figure shows the ShiftCrypt profile for the bounded (orange) and free (blue) states. The upper part shows its relation with the structure: the RimM (in green) in complex with the ribosomal protein S19 (in orange). The five blue residues are the ones that are reported to change their ShiftCrypt value the most. Since in these proteins there are several missing values, the analysis has been carried out using only the most common atoms (H, $C^{\alpha}$, and N) in order to calculate the ShiftCrypt index

avoiding the use of "secondary shifts", which seem to contain conformational biases[17]. It so enables an easier and more accurate comparative analysis between different residues or proteins.

The correlation of the ShiftCrypt index to both structural and non-structural biophysical properties that are essential for native protein behavior, without being explicitly linked to any of them, indicates that it can be considered to be an encoded biophysical feature that describes the in-solution behavior of amino acids in solution. It so provides an advantage for data analysis, with researchers not having to combine or visualize higher-dimensional data. ShiftCrypt can be used to find chemical shift-based similarities and differences in the biophysical behavior of different proteins, or between different states of the same protein, without having to employ heuristic interpretations of the chemical shift data. This is also useful where structure models have already been calculated, as especially dynamical in-solution information is inevitably lost during the structure calculation process. Furthermore, the ShiftCrypt method is flexible and can

be adapted to particular problems while accounting for the chemical shift data that are typically available in a given situation. ShiftCrypt may be used, for example, to find fragment similarities for CS homology-based structure calculation of NMR structures, or for the amino acid type-independent identification of peptide fragments in databases.

## Methods

**Dataset**. The dataset to train the method is composed of 3385 NMR structures annotated with chemical shift values and re-referenced by the structure-based VASCO method[18]. In order to remove misreferenced atoms, we filtered out of the dataset all the residues that contained atoms with extreme values (< of percentile 1 or > of percentile 99). For the annotation of the secondary structure of each amino acid, we evaluated all structure models in each NMR ensemble with STRIDE[19], and considered as helix, sheet, or coil only those residues that were consistently assigned to that type of secondary structure throughout the ensemble. These structures have only been used to relate ShiftCrypt to the structural characteristics of proteins and have not been used in the training procedure. For the dimer validation, we downloaded the CS of the four complexes from the BMRB database[2].

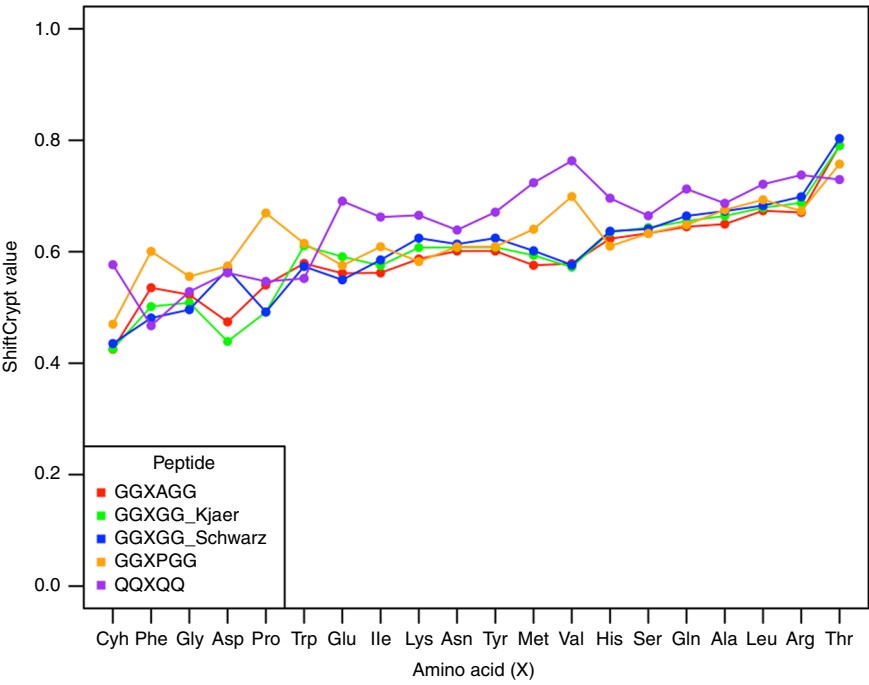

**Fig. 6** ShiftCrypt index for a series of random coil peptides. The amino acids are ordered by increasing average ShiftCrypt value over all peptides. The GGXPGG and GGXAGG peptides come from ref. [16], the QQXQQ from ref. [17], the GGXGG schwarz, and the GGXGG Kjaer from refs. [15,21], respectively

We discarded the residues with missing values in the N, H, or Cα atoms and we obtained a dataset of 280 residues.

**Data extraction encoding scheme definition.** For each amino acid type, we selected a set of atoms that are taken into consideration during the ShiftCrypt definition. This selection is based exclusively on the relative data abundance in our training dataset. The atoms taken into consideration for each amino acid type are listed in Supplementary Table 5. In addition to the model that uses the full list CS that are usually obtained in complete NMR experiments, we provide two supplementary models: the first one uses only the most commonly analyzed atoms: in this model, we use only the N, C, Cα, Cβ, H, and Hα atoms, and it is suitable to deal with proteins with missing data. The second one uses only the CS of N and H atoms of the amino group and the Cα. This model has been used to perform the analysis of the dimers and it overcomes the large amount of missing data. We provide the code of ShiftCrypt training scripts and training data that can be used to easily build a custom encoding scheme.

**Data scaling.** In order to correctly train the neural network, CS are required to undergo an amino acid type-specific scaling. This is done by simply scaling the data in a residue-specific way between 0 and 1 using the MinMaxScaler from the scikit-learn python library[20]. The scaled data are obtained as $\mathbf{X}_{\text{scaled}} = (\mathbf{X} - \min(\mathbf{X}))/(\max(\mathbf{X}) - \min(\mathbf{X}))$, where $\mathbf{X}$ is the original data and $\mathbf{X}_{\text{scaled}}$ the scaled one. Moreover, this procedure helps in reducing the differences between chemical shift values obtained from different amino acid types.

**Autoencoder transformation.** In order to extract the maximum amount of information from the chemical shift data, we built a feed-forward autoencoder for each amino acid type, using the python library PyTorch (https://pytorch.org). The model is divided into two parts (see Supplementary Fig. 17), which are identical and mirrored: the encoder and the decoder. The encoder takes as input the list of scaled CS of a residue and passes it through two hidden layers of 100 neurons, each with rectified linear unit (ReLU) activations. The last layer is made of a single neuron with sigmoid activation. The decoder performs the reverse operation by taking the output of the last encoder neuron, and by passing through the two hidden layers, it aims to replicate the input CS. The compressed chemical shift information is the value of the sigmoid neuron. It is important to note that, since the sigmoid activation function is bijective, the decoding network will always map the compressed information to an unambiguous set of simplified chemical shift values. The list of per-atom Pearson's correlation coefficients between the actual CS and the simplified CS is available in Supplementary Data 1. In order to make the ShiftCrypt easier to understand, in the training procedure, we make sure to invert the direction of the index for the residue types that show a high beta-sheet propensity at low ShiftCrypt values and a high helix propensity at high ones. This can be safely done, since the direction in which the neural network learns the chemical shift information is

arbitrary (see the "Method_Explanation.pdf" file of the ShiftCrypt repository for a practical illustration of the learning procedure of the neural network).

**Stability of the model.** Neural networks are nonlinear models, and due to their complexity, small changes in their parameters may lead to strong changes in the way in which information is extracted from data. This is usually noticeable in deep and ultra-deep neural networks. Since ShiftCrypt is made of just three layers of neurons, this effect is expected to be very small. However, in order to explore the stability of the model, we tested if the relationship between secondary structure propensity and ShiftCrypt index was maintained when varying the number of hidden neurons of the network. We tested 15 different models, with a number of hidden neurons per layer, that varied between 10 and 150. Supplementary Figs. 18 and 19 show that the median ShiftCrypt value of each secondary structure population varies very little with the number of hidden neurons of the neural network, even if we consider every single amino acid separately.

## Data availability
All the data used in this paper are available at https://bitbucket.org/bio2byte/shiftcrypt.

## Code availability
A Python implementation of the algorithm is available at https://bitbucket.org/bio2byte/shiftcrypt. It can also be installed via the shiftcrypt PyPI package.

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

## Acknowledgements

G.O. and D.R. are grateful to A. Motz, A.L. Mascagni for the support and the helpful discussions, and to R. Gilbert and R. Sanchez for the inspiration. D.R. is funded by a Research Foundation—Flanders (FWO) Post-Doctoral Fellowship. G.O. acknowledges funding by the Research Foundation Flanders (FWO) - project nr. G.0328.16N.

## Author contributions

G.O. developed the algorithm. G.O. and W.V. performed tests and validations. G.O., W.V. and D.R. wrote the article.

## Additional information

**Competing interests:** The authors declare no competing interests.

