## [Peer Review File · Nature Communications]

Reviewers' comments:

Reviewer #1 (Remarks to the Author):

The manuscript describes a neural network approach to unify protein chemical shifts and their changes in response to changes in protein sequence, structure, and dimerization interactions. An argument can be made that such a unification has certain benefits over alternative chemical analysis methods that are currently in use. Unfortunately, the authors fail to provide convincing comparisons of their new ShiftCrypt Index with other metrics, such as the secondary structure propensity index (SSP) and direct chemical shift perturbation. Therefore, the authors' case for their new index is not convincing.

Specific comments:

1. Figure 3: the two homologous proteins have very similar secondary structures, which would also be reflected in their SSPs. How do they compare? Is the small difference in the ShiftCrypt index at the end of a b-strand significant? It appears that there are other structural differences between the two structures of similar magnitude. Why don't they also have different ShiftCrypt indices?
2. Figure 4: similar to 1., how do the SSP compare between the two proteins? Also, what is the significance and origin of the large difference in the ShiftCrypt index around residue 20? A careful analysis of the false positive rate of the ShiftCrypt index needs to be presented.
3. Figure 5: how is the dimerization interface reflected by a simple chemical shift perturbation analysis of ^{15}N - ^1H HSQC peaks? Is there any added value of ShiftCrypt?
4. The nomenclature of different indices in text and figures is inconsistent and needs to be straightened out.

Reviewer #2 (Remarks to the Author):

NMR chemical shift information is highly valuable in many applications that deal with molecular structures, as it couples strongly to structural and chemical features of the close environment of the atom. Significant research efforts have been invested into understanding and simulating such chemical shifts.

In this work, the authors present an approach based on artificial neural networks to develop an alternative measure, based on the chemical shift, that is easier to interpret and use in practical applications. The measure, called ShiftCrypt by the authors, is derived by using an auto encoder on all chemical shifts of a given residue which compresses them into a single number. This is a sound and modern approach towards this problem.

The idea behind this approach is to find a single quantity that contains sufficient information to reproduce the individual shift values of the atoms and, hence, describes an intrinsic hidden feature that is simpler to interpret and can be used more generally.

The resulting index is shown to correlate well with certain features of interest (such as secondary structure elements). The authors also demonstrate that it yields a useful property in protein comparison.

The method is, in my opinion, very interesting and clearly methodologically sound and innovative. It has the potential to improve several structural applications, and the results do look convincing.

However, while the evaluation is thorough, I find it very hard to quantify the true advantages of the score compared to using the shift information directly. Here, the manuscript could profit from more quantitative comparison to methods based on chemical shifts, and methods using the new ShiftCrypt index.

We thank both reviewers for their comments, which enabled us to significantly revise and clarify the work described in the manuscript. Please note that in addition to addressing the comments of reviewers below, we modified some parts of the text and rebuilt some of the figures in order to make the paper clearer and easier to understand. First, we modified the indexing of the sequences in Figures 3, 4 and 5 in order to take into consideration gaps and missing values. Secondly, for Figure 5 we used a model that requires much less chemical shift information in order to overcome the problem of missing data in the dimer chemical shifts, resulting in a new statistical analysis described in section 2.5.

Reviewer #1 (Remarks to the Author):

The manuscript describes a neural network approach to unify protein chemical shifts and their changes in response to changes in protein sequence, structure, and dimerization interactions. An argument can be made that such a unification has certain benefits over alternative chemical analysis methods that are currently in use. Unfortunately, the authors fail to provide convincing comparisons of their new ShiftCrypt Index with other metrics, such as the secondary structure propensity index (SSP) and direct chemical shift perturbation. Therefore, the authors' case for their new index is not convincing.

We have further clarified the information provided by the ShiftCrypt index, and added more extensive comparisons with methods that derive secondary structure content from chemical shifts such as d2D and SSP, both for proteins with similar structure, and for changes in a protein upon complex formation. The direct chemical shift differences between two distinct states of the same protein do in principle provide information similar to ShiftCrypt, but the question of how to combine or interpret the multidimensional vector representing these chemical shift differences remains. In such cases, ShiftCrypt provides a more direct means to assess the differences in biophysical behavior encoded by the chemical shifts of the two distinct protein states. In addition, we have added an analysis of GGXGG-type random coil peptides, which provides a further perspective on the ShiftCrypt index. It illustrates that, when comparing the chemical shifts of proteins with a different sequence, the use of secondary chemical shifts might not accurately reflect the actual behavior of amino acid residues in solution.

Specific comments:

1. Figure 3: the two homologous proteins have very similar secondary structures, which would also be reflected in their SSPs. How do they compare?

With Figures 3 and 4 we want to show that ShiftCrypt extracts information from chemical shift data that is coherent with what is known about these proteins. The current approach to aligning similar structures is in essence through the superimposition of their (similar) secondary structure elements, with protein alignment benchmarks also built on this assumption. We therefore chose, for Figures 3 and 4, two pairs of proteins with extremely well conserved structures,

where there is little question about the position of the secondary structure elements. However, these structures are 'snapshot' models derived from experimental data, with highly precise atomic positions that do not reflect the full behavior of the protein in solution. Relatively small changes in the secondary structure, as observed in the model, may therefore reflect much larger changes in the actual solution behavior of the protein. Conversely, the model may be similar but the actual solution behavior may be different, or the models may differ slightly because of experimental data reasons, with the solution behavior similar. The key point is that it is not evident to directly derive from small conformational differences in the model what is actually happening in solution. We have also added a more thorough analysis of the NMR ensembles of these proteins in the supplementary data.

There is, in any case, no question about the presence and position of the secondary structure elements, and as demonstrated in Figures S9-S13, the SSPs as determined by d2d and the SSP index are indeed very similar, with some differences in total content. The RCI profile, on the other hand, is quite dissimilar. Compared to the ShiftCrypt profiles, the secondary structure propensity profiles remain very similar overall, with more conserved 'fine' structure being present at the residue level for ShiftCrypt, even where the amino acid changes. In addition, the first beta-strand is similar to the other beta-strands according to the secondary structure propensity methods, whereas it has different (lower) values according to the ShiftCrypt index. This likely reflects a difference in the in-solution behavior of the first beta-strand, possibly because of its N-terminal position. Interestingly, in the structure analysis this part is not well defined in the *A. thaliana* thioredoxin, but is in the *S. cerevisiae* one (Figure S16), while their ShiftCrypt indexes are similar, indicating that one of the structures might be over- (or under-)defined. These examples illustrate that the methods are therefore not equivalent with respect to the information they pick up from the chemical shift data.

ShiftCrypt is, crucially, not a predictor or an homology detector, but an agnostic chemical shift data compressor, of which the resulting value reflects the biophysical properties of the amino acid in solution. The secondary structure correlations are a side effect of this compression, and as such ShiftCrypt cannot compete with secondary structure propensity methods for the singular identification of secondary structure propensities, but that is also not its aim, as ShiftCrypt will provide information for any kind of amino acid, well-structured or flexible. Importantly, ShiftCrypt does not rely on using a set of reference chemical shift values, which have their own issues (see newly added section 2.6 about random coil peptides). This also means ShiftCrypt could, for example, be used to find similarities between IDPs in solution, as it goes beyond secondary structure content. Unfortunately, we could not identify any data on IDPs that we can use to test this claim, and therefore limit the analysis here to proteins with clearly determined and conserved secondary structure elements. We modified the text in the manuscript in order to make the above points clearer.

Is the small difference in the ShiftCrypt index at the end of a b-strand significant?

We added a statistical analysis in the text, based on the cumulative distribution F of the observed differences between the profiles of the two allergenes. We selected and analyzed the portions of the proteins in which the shiftCrypt index was significantly discordant ($F(x) > 0.95$ fitting the differences with a Gaussian distribution). We therefore selected positions 38, 149, 72, 88, 122 and 77, showing that they correspond to regions that are likely to behave differently in solution.

It appears that there are other structural differences between the two structures of similar magnitude. Why don't they also have different ShiftCrypt indices?

Single, static models of the proteins' conformation are often not a good approximation of some events that might happen in solution. Structure ensembles calculated from NMR data provide somewhat better information, but are still limited in that they model the (conformationally averaged) in solution NMR data to single models, typically using an iterative process of NOE peak assignment that can result in overdefined structures. The Supplementary data analysis on the thioredoxins (Figure S15) reflects this. The point that we want to make is that the ShiftCrypt index reflects the in-solution biophysical behavior of amino acids in the protein, so providing a view on the protein that can be 'lost' in the conversion of the NMR data into (multiple) single models.

2. Figure 4: similar to 1., how do the SSP compare between the two proteins?

We have included this in the earlier discussion.

Also, what is the significance and origin of the large difference in the ShiftCrypt index around residue 20?

The difference around position 20 (now around position 30 with the new indexing that includes gaps and missing values) corresponds to a loop between the start of a beta sheet (on one side) and the end of an alpha helix (on the other). The beta sheet is shorter in the yeast protein, suggesting the two proteins may have different in solution behavior. For what concerns the differences in the very first part of the protein (position 4-8) they are just next to the active site, suggesting they may be caused by organism-specific structural differences.

A careful analysis of the false positive rate of the ShiftCrypt index needs to be presented.

ShiftCrypt is not a predictor, and we do not have a ground truth to compare to. Our method provides a single value derived from the experimental chemical shift information, which is interpretable in terms of classifications such as secondary structure, but this is not the aim of the ShiftCrypt index, and would in our view limit its use.

A more quantitative evaluation of our method could be related to homology detection, but this is challenging since chemical shifts refer to in-solution behavior, while homology is often defined by the superimposition of two static conformations. We therefore agree that our manuscript lacks a clear quantization of the advantages of our tool, which we have attempted clarify further in the analysis in section 2.5 and in point 3 of our answers here. Moreover, we performed an analysis of GGXGG-type random coil peptides that are usually used as “standard” for many tools, but that have been recently shown to have different in-solution behavior (see section 2.6).

3. Figure 5: how is the dimerization interface reflected by a simple chemical shift perturbation analysis of ^{15}N - ^1H HSQC peaks? Is there any added value of ShiftCrypt?

In order to address this point, we analyzed the ability to detect interaction patches from the chemical shifts of 280 residues, we built an ad-hoc ShiftCrypt model that uses only the CO, N and HN atoms as input. We show that this modified ShiftCrypt is able to distinguish residues that interact from ones that do not, while the secondary structure content methods fail to do so. Moreover, we find that the ShiftCrypt provides orthogonal information with respect to the existing ^{15}N - ^1H *ad-hoc* method for the detection of interacting residues, as the difference between interaction and non-interaction patches significantly increases when these approaches are combined.

We performed this more quantitative analysis also in accordance with the comments of reviewer #2, in order to provide a more accurate comparison with state of the art chemical shifts-based tools and typical methodologies. We took 4 protein dimers for which we were able to retrieve the CS data for both the free and bound form, for a total of 280 residues. We annotated every residue that was part of the interaction patch. We then ran different tools on the free and bounded pairs (the 3 secondary structure propensities of d2D, PSS, ShiftCrypt and the ^{15}N - ^1H HSQC method described in Williamson 2013). We then calculated the per-residue difference of the various indices for the free and bound form. Intuitively, if the index is capable of identify the residues in the interaction patches, the differences should be higher for the amino acids annotated as part of the patch respect to the others. We performed a Wilcoxon rank sum test for all the 7 indices in order to test which ones were associated with higher differences in the interaction patches. Interestingly, the only significant results were ShiftCrypt and the ^{15}N - ^1H HSQC method that you suggested (p-value 10^{-7} for both of them). This confirms the well-known point that, even if there is no change in the secondary structure propensities, chemical shifts still contain information about dimerization, especially through hydrogen bond changes which affect the backbone NH and CO chemical shift values.

While the performance in distinguishing interaction residues is basically the same for ShiftCrypt and the ^{15}N - ^1H HSQC method, the Pearson's correlation coefficient between the differences highlighted by these two methods is only

0.58, decreasing to 0.38 if we only consider the residues in the interaction patches. This means that, although both approaches detect these changes, they provide two complementary sources of information. Note that this is maybe not surprising given that the ShiftCrypt index includes CO values, but as pointed out by Williamson these are considered to be less useful or at least more difficult to interpret in the traditional chemical shift interpretation context.

To further verify the synergy between the two methods, we built a naive “consensus” score, by simply summing the combined ^{15}N - ^1H HSQC chemical shift change and the ShiftCrypt index. The resulting p-value of this “mixed” score is reduced to 10^{-10} . This shows that the two methods are complementary and that a more complex machine learning consensus method may provide even better ways to estimate the patches of interaction. Note that Williamson points out the difficulty in setting the scaling factor between ^{15}N and ^1H in the *ad hoc* method, which is not an issue for ShiftCrypt. In addition, ShiftCrypt could in principle deal with amino acid mutations as long as the monomer behavior is not significantly affected. A final issue to consider is that the ^{15}N - ^1H HSQC chemical shift change was likely used in the structure calculation of these complexes in methods such as HADDOCK, so biasing their correlation.

The key point in this context is that the ShiftCrypt approach can be incorporated and tailored to specific uses of chemical shift information. In the final git repository we will provide the source code of the training script that can be used to build ShiftCrypt with different encoding schemes, enabling users to deploy the method in specific real case applications.

Additionally, in accordance with the comments about Figure 3 and 4, we modified Figure by visualising the full protein sequence in the plot, and by highlighting the 5 residues in which the ShiftCrypt value changes the most.

4. The nomenclature of different indices in text and figures is inconsistent and needs to be straightened out.

Thank you, we have modified this.

Reviewer #2 (Remarks to the Author):

NMR chemical shift information is highly valuable in many applications that deal with molecular structures, as it couples strongly to structural and chemical features of the close environment of the atom. Significant research efforts have been invested into understanding and simulating such chemical shifts.

In this work, the authors present an approach based on artificial neural networks to develop an alternative measure, based on the chemical shift, that is easier to interpret and use in practical applications. The measure, called ShiftCrypt by the authors, is derived by using an auto encoder on all chemical shifts of a given residue which compresses them into a single number. This is a sound and modern approach towards this problem.

The idea behind this approach is to find a single quantity that contains sufficient information to reproduce the individual shift values of the atoms and, hence, describes an intrinsic hidden feature that is simpler to interpret and can be used more generally.

The resulting index is shown to correlate well with certain features of interest (such as secondary structure elements). The authors also demonstrate that it yields a useful property in protein comparison.

The method is, in my opinion, very interesting and clearly methodologically sound and innovative. It has the potential to improve several structural applications, and the results do look convincing.

However, while the evaluation is thorough, I find it very hard to quantify the true advantages of the score compared to using the shift information directly. Here, the manuscript could profit from more quantitative comparison to methods based on chemical shifts, and methods using the new ShiftCrypt index.

We have addressed these comments in the answers to the first reviewer in parts 1-3.

Editorial Note: Reviewer #1 was unable during the second round of review. Reviewer #3 was recruited as a substitute.

Reviewers' comments:

Reviewer #2 (Remarks to the Author):

The revision of the manuscript has greatly improved the paper in my opinion. The rationale behind the ShiftCrypt index is now clearly stated in the text. The additional comparisons and discussions help in establishing that ShiftCrypt is an additional useful source of information.

Obviously, it would be desirable if there was a more quantitative way of showing the superiority of the newly proposed index over well-known ones. On the other hand, it is not really a matter of superiority - the important question is whether this measure yields additional useful information that is not trivially related to well-known quantities. In my opinion, the authors have clearly demonstrated that this holds, and hence, that ShiftCrypt will be a highly useful addition to the toolbox of computational shift analysis.

Reviewer #3 (Remarks to the Author):

This revised manuscript, which I have not reviewed previously, presents a method to compress the chemical shift data of amino acid residues into a single number between 0 and 1, the ShiftCrypt index (SCI). The paper presents various applications of the SCI but is very terse (less than one page in the Methods section) on how it is defined and calculated, and I think there are some fundamental issues with that, which should be addressed before publishing the paper.

1. The paper would be easier to understand if the definition and computation method of the SCI were given first, before presenting applications. The way the paper is written now, the applications try to ascribe almost miraculous properties to a black box quantity. Properties of the SCI should be discussed more in depth/fundamentally (see below) than in the present form of the manuscript.

2. The SCI is determined by a symmetric neural network that takes as input the (somehow normalized) chemical shift values, transforms it into a single number (the SCI), and then tries to reproduce the individual chemical shift values by an (identical?) inverted neural network. The latter is needed to enable training based on known chemical shifts. Obviously (as also pointed out by the authors on p.14) any bijective transformation of the interval [0,1] applied to the output value of the single central neuron will produce another SCI* that may take totally different values for the same input chemical shifts but has an equivalent information content as the original SCI. This means that reporting "absolute" values of the SCI is meaningless (without complete specification of the neural network that computes them).

3. Therefore, I think that the SCI can only be used to detect differences between closely related proteins, in the same way as comparing (appropriately scaled and combined) chemical shift changes, not as a meaningful generic feature of one protein. The applications seem to confirm that.

4. If I understand correctly, the neural network is trained independently for each amino acid type, possibly using chemical shifts from different atoms. Since, as pointed out above, there are infinitely many different, but essentially equivalent ways to define an SCI, I don't understand how this procedure will yield a "residue independent" SCI that can be compared between different residue types, especially if for some predefined residue types the SCI is replaced ad hoc by 1 - SCI (p. 14). Even for a given residue type, one would expect that training different instances of the neural network with the same data would yield different SCIs from the family of equivalent ones. Is this true, or being avoided by some sort of biasing?

5. How similar are the output chemical shift values of the neural network to the input ones? Correlation plots should be included in SI and it should be reported for which fraction of the chemical shifts input and output values agree within a reasonably small tolerance (e.g. one half or one third of the standard deviation in the BMRB statistics).

6. Obviously, points in an n-dimensional space can always be mapped to a 1-dimensional number

with approximate but arbitrarily accurate “bijectivity” by dividing up the space into (more or less cleverly) indexed cells. How does the neural network-based SCI compare with such trivial approaches?

7. Have the proteins that are used in the example applications been excluded from the training set?

8. The choice of atoms included in the full atoms encoding scheme deserves some explanation: Are reduced/oxidized Cys considered together or separately? Why are the Val methyls and the Ile delta1 methyl group included but not the Leu methyls? Why are methyl protons included but not methyl carbons? For Val, HG22 and HG23 are included but not the third proton of the methyl group, HG21? Are the three protons of a methyl group that always have the same chemical shift treated as a single entity? Why are all aromatics excluded?

Details:

- p 6: “RMSD of 1.756” lacks the unit (Å).
- Fig. 3. Caption should explain what is green and brown, and the red vertical lines in the plot.
- p. 13: The statement that the SCI “does not require heuristic formulas” is in my opinion not justified: The design and parameters of the neural network include also “heuristic” components.
- p. 14: The scaling mentioned in the last sentence should be described more specifically.
- p. 14: Ref. 1 lacks issue/page information.

1. The paper would be easier to understand if the definition and computation method of the SCI were given first, before presenting applications. The way the paper is written now, the applications try to ascribe almost miraculous properties to a black box quantity. Properties of the SCI should be discussed more in depth/fundamentally (see below) than in the present form of the manuscript.

We moved some of the method description to the introduction and we added more details about the method implementation in the Methods section. We also reorganized the methods section itself.

With regard to the results, we are only describing what we observe when applying the method. We think ShiftCrypt works because, despite the complexity of multidimensional chemical shift space, there are dependencies between the chemical shift values that are present in all amino acids.

2. The SCI is determined by a symmetric neural network that takes as input the (somehow normalized) chemical shift values, transforms it into a single number (the SCI), and then tries to reproduce the individual chemical shift values by an (identical?) inverted neural network. The latter is needed to enable training based on known chemical shifts. Obviously (as also pointed out by the authors on p.14) any bijective transformation of the interval [0,1] applied to the output value of the single central neuron will produce another SCI* that may take totally different values for the same input chemical shifts but has an equivalent information content as the original SCI. This means that reporting “absolute” values of the SCI is meaningless (without complete specification of the neural network that computes them).

The approach we used is an example of NN-based autoencoder, which is generally used for dimensionality reduction through the unsupervised learning of an efficient low-dimensional encoding of high-dimensional data. The two parts of the network before and after the “bottleneck neuron” are indeed identical. Among all the possible transformations, which are indeed almost infinite, the neural network is pushed to choose an “optimal” mapping that maximizes the information content of the “bottleneck neuron” encoding. One way in which the network can do that is by assigning similar representations to similar sets of chemical shifts, thus maximizing the information “density” on the sigmoid. At the protein level, similar sets of chemical shifts tend to have similar “structural behavior” and this is indeed what we observe in the plots of the secondary structure and backbone dihedral angles. We agree that the neural network may, in principle, find different “mappings” for the input chemical shifts to the output sigmoid neuron (SCI), but the probability of convergence to (almost) the same solution is very high for a shallow network like this. Our network has only 2 hidden layers and consistently reaches convergence after just 5-10 epochs. Also the analysis of the data show a clear commonality between different amino acid types, indicating this is not an issue.

In order to prove that the different models converge to similar solutions, we analyzed how the relation between secondary structure propensity and SCI changes by modifying the parameters of the neural network. In particular, we can demonstrate that the median SCI value of each secondary structure population basically does not change when varying the number of hidden neurons of the neural network (Supplementary Figure 18). This means that different models converge to very similar solutions that maintain the absolute SCI properties described in the paper.

We also included a new section in the manuscript about the stability of the model.

3. Therefore, I think that the SCI can only be used to detect differences between closely related proteins, in the same way as comparing (appropriately scaled and combined) chemical shift changes, not as a meaningful generic feature of one protein. The applications seem to confirm that.

As mentioned above, the neural network is pushed to map similar sets of chemical shifts with similar values of the sigmoid. As we show in Figures 1-2 and Supplementary Figures 1-8, as well as in the new Supplementary Figures 18 and 19, this is also what we observe when we apply the method. The method does pick up similarities between proteins with very different sequences, and therefore chemical shift values for different amino acids in the same sequence position (Figures 3 and 4).

4. If I understand correctly, the neural network is trained independently for each amino acid type, possibly using chemical shifts from different atoms. Since, as pointed out above, there are infinitely many different, but essentially equivalent ways to define an SCI, I don't understand how this procedure will yield a "residue independent" SCI that can be compared between different residue types, especially if for some predefined residue types the SCI is replaced ad hoc by 1 - SCI (p. 14). Even for a given residue type, one would expect that training different instances of the neural network with the same data would yield different SCIs from the family of equivalent ones. Is this true, or being avoided by some sort of biasing?

ShiftCrypt is not a deep neural network, as it has only 3 layers. The effective complexity of the mappings that it can learn is thus limited by this constraint. Moreover, we repeated many times the training procedure to check its stability, and it always converges to similar mappings in every of the hundreds of experiments we ran (see also point 2). This is also verifiable by using the training scripts and dataset provided in the git repository.

The concept behind ShiftCrypt is based on the assumption that chemical shifts are not distributed randomly in their vector space, due to the physico-chemical restraints that amino acid residues have, especially in proteins. In other words, the chemical shifts are distributed over a manifold-like subspace that is influenced by the structural and chemical properties of the amino acid residues. If this was not the case, the auto-encoder would fail to summarize the chemical shifts. This is also illustrated by the analyses shown in Figures 1-2 and Supplementary Figures 1-8. The inherent residue independence of the ShiftCrypt index, which goes beyond current methods, is likely due to the chemical shifts representing the general behavior of a residue in solution, so that basically every residue chemical shift set encodes the same type of information, but expressed in different way. The NN homogenizes this while, as shown in our analysis, maintaining the general structural behavior.

We included an additional figure (Supplementary Figure 19) in which we show that the median SCI value of each secondary structure population varies very little with the number of hidden neurons of the neural network, even if we take every single amino acid separately. This also shows how varying the model we still get similar mappings and how even very different (non-deep) neural networks converge to similar solutions. We also included a detailed explanation of the principle behind ShiftCrypt in the README of the git repository.

5. How similar are the output chemical shift values of the neural network to the input ones? Correlation plots should be included in SI and it should be reported for which fraction of the chemical shifts input and output values agree within a reasonably small tolerance (*e.g.* one half or one third of the standard deviation in the BMRB statistics).

This strongly depends on the group of atoms that is taken into consideration. We now included the Pearson's correlation coefficients (PCC) of each atom as a supplementary text file. Moreover, the PCCs are shown at screen at every re-train of the model. Please take into consideration that, even if an atom has a low PCC, it may still help in increasing the resolution of other atoms. In other words, a PCC close to 0 does not mean the atom is useless in the encoding scheme.

6. Obviously, points in an n-dimensional space can always be mapped to a 1-dimensional number with approximate but arbitrarily accurate “bijection” by dividing up the space into (more or less cleverly) indexed cells. How does the neural network-based SCI compare with such trivial approaches?

There are an infinite number of transformations that map chemical shifts to a lower dimensional space, for example d2D and PSS. We want to avoid “benchmarking” such possible transformations, because: 1) they are infinite and 2) we think the definition of a “best transformation” is not necessarily meaningful and will necessarily depend on the (subjective) evaluation criteria and target (e.g. dynamics, conformation) that are adopted. With our approach we want to minimize any bias.

7. Have the proteins that are used in the example applications been excluded from the training set?

Yes, they were excluded from the training set. We have clarified this in the manuscript.

8. The choice of atoms included in the full atoms encoding scheme deserves some explanation: Are reduced/oxidized Cys considered together or separately?

They are considered together at the moment. We are aware that there are methods for determining the oxidation state of cysteines based on their chemical shifts, but as we here want to demonstrate a novel concept, and keep it as free from bias as possible, we want it to remain agnostic about existing chemical-shift based methods. The oxidation state of the cysteines is likely the reason that there is, for example, more overlap between the secondary structure categories for Cys (see supplementary Figure 2). It also attests to the robustness of our method that it works despite this variation in the input.

Why are the Val methyls and the Ile delta1 methyl group included but not the Leu methyls? Why are methyl protons included but not methyl carbons? For Val, HG22 and HG23 are included but not the third proton of the methyl group, HG21? Are the three protons of a methyl group that always have the same chemical shift treated as a single entity? Why are all aromatics excluded?

The choice of the atoms to include in the model has been performed exclusively on the basis of the amount of data available in our training set. We provide, along with the models used in the paper, the training scripts to 1) retrain the SCI on different training sets 2) change the encoding scheme of each residue, so including/excluding different atoms. The script is intended to be easy to use even for people without a strong computational background.

The current ShiftCrypt implementation can certainly be refined for practical and pragmatic purposes, and we intend to work on this in the future, in particular on documentation.

Details:

- p 6: “RMSD of 1.756” lacks the unit (Å).

Done, thanks

- Fig. 3. Caption should explain what is green and brown, and the red vertical lines in the plot.

Done, thanks

- p. 13: The statement that the SCI “does not require heuristic formulas” is in my opinion not justified: The design and parameters of the neural network include also “heuristic” components.

Neural networks are indeed intrinsically heuristic method due to the fact the solution is obtained by minimizing a loss function using heuristic optimizers. On the other hand, with this sentence we mean that what we observe is what comes directly from the data, without any manual, *a priori* or *a posteriori*, correction or interpretation. As for the hyper-parameter optimization, we showed that model is very stable. To avoid any misunderstanding, however, we removed the sentence.

- p. 14: The scaling mentioned in the last sentence should be described more specifically.

The MinMaxScaler is a commonly used scaling transformation used in machine learning. It translates the input between 0 and 1 with the formula $X_{\text{scaled}} = (X - \min(X)) / (\max(X) - \min(X))$. We added the formula to the text.

- p. 14: Ref. 1 lacks issue/page information.

Done, thanks

Reviewers' comments:

Reviewer #3 (Remarks to the Author):

The authors have responded to the questions and remarks of the reviewer.

I recommend publication after revision to address the following points.

p. 14, Data Scaling: This scaling, which corresponds to the well-known approach of using secondary chemical shifts, is probably the main origin of the residue type independence of the ShiftCrypt Index, rather than the use of neural networks. This should be stated.

p. 14, Autoencoder Transformation: Given the stability of the median ShiftCrypt Index values of different residue types for different secondary structure types, it is still unclear why the direction of the ShiftCrypt Index must be inverted ($x \rightarrow 1 - x$) for certain residue types. This should be explained.

Supplementary Table 5/File pearsons.txt: The choice of atoms does not make sense for methyl groups. The 3 hydrogens of a methyl group have ALWAYS the same, degenerate chemical shift value. This contradicts the observation that the correlation coefficients in pearsons.txt are different for them. Sometimes only 2 out of 3 methyl hydrogens are listed (e.g. Ala HB*, Val HG2*). Leu methyls are commonly observed yet not included.

Supplementary Table 5: Met & Asn NB2 should probably be HB2?

File pearsons.txt: Does the lack of significant correlation between input and predicted chemical shift values for most atoms except CA, CB, C, HA show that individual shift values cannot be recovered from the 1D ShiftCrypt Index but in general the right region in the multidimensional manifold of populated shifts is found?

File pearsons.txt: Data is repeated multiple times?

Supplementary Fig. 19: Scale for the y-axis missing? Identical for all panels?

Method_Explanation.pdf: Explain the mathematical relation between the ShiftCrypt Index and the black lines in the 2nd and 3rd Figures. Is the last paragraph the answer to my question above on inverting the direction of the ShiftCrypt Index for certain residue types? If so, why does the neural network intrinsically "switch to inverted values" for these residues?

p. 14, Data Scaling: This scaling, which corresponds to the well-known approach of using secondary chemical shifts, is probably the main origin of the residue type independence of the ShiftCrypt Index, rather than the use of neural networks. This should be stated.

We agree that scaling reduces the differences between residue types and we now state this in the text. However, as shown in figure 4a and 4b, single scaled (or secondary) chemical shifts are less conserved in the homologous proteins with respect to the ShiftCrypt score.

In addition, using Neural networks without scaling the data is in most of cases considered a bad practice that may lead to sub-optimal results. Activation functions are indeed designed to work on a limited range of values, in order to guarantee that the gradients are correctly propagated through the network during the training phase. Extreme values may force the optimizer to find sub-optimal solutions, leading to the effective “death” of specific neurons (*e.g.* the back-propagated gradient is 0 and thus no optimization of the downstream neurons can be performed). For example, very high or low values with a sigmoid activation function would flatten the gradient function, making the optimizer unable to find the direction in which the weights should be pushed. Another example is the $x < 0$ region in ReLU functions.

p. 14, Autoencoder Transformation: Given the stability of the median ShiftCrypt Index values of different residue types for different secondary structure types, it is still unclear why the direction of the ShiftCrypt Index must be inverted ($x \rightarrow 1 - x$) for certain residue types. This should be explained.

The direction is arbitrary for all the residue types. This is due to the fact that the neural network is not trained as a classifier, but as an auto-encoder. For example, let us imagine that we have an auto-encoder with only linear activations and no biases, which, once trained, produces output that corresponds to the ShiftCrypt Index. If we take the weights of an internal layer of the encoder and we multiply them by -1, doing the same for the correspondent layer in the decoder, the output of the encoding-decoding procedure will be exactly the same. In this second case, the encoded value will have an inverted direction respect to the previous one. The choice of the direction is in practice done randomly by the optimizer, in function of the initialization values of the weights and of the optimization algorithm. The two solutions have nonetheless the same loss value. The same concept can be applied to NN with different activation functions and a bias. We take the solution that shows higher helix propensity on the left side to make the index easier to understand and to interpret for the users. We added this example in the “Readme” of the repository.

Supplementary Table 5/File pearsons.txt: The choice of atoms does not make sense for methyl groups. The 3 hydrogens of a methyl group have ALWAYS the same, degenerate chemical shift value. This contradicts the observation that the correlation coefficients in pearsons.txt are different for them. Sometimes only 2 out of 3 methyl hydrogens are listed (*e.g.* Ala HB*, Val HG2*). Leu methyls are commonly observed yet not included.

The atom naming in this table was indeed confusing; we have cleaned up nomenclature and updated the encoding scheme accordingly in Table S4 and the Figures.

The results do not change, which we think is due to the main backbone atoms being the strongest determinants of the encoding, with the side-chain atoms are less important, so they do not strongly influence the output of the network.

The key issue here, however, is to allow users the flexibility to experiment and adapt the model to their requirements; we so provide an easy way to retrain the ShiftCrypt Index and obtain custom models with different encoding schemes.

The fact that the neural network provided different estimated chemical shifts for redundant atoms is that NNs are trained by heuristic optimizers and the solution is always an approximation.

Supplementary Table 5: Met & Asn NB2 should probably be HB2?

Yes, thanks for noticing.

File pearsons.txt: Does the lack of significant correlation between input and predicted chemical shift values for most atoms except CA, CB, C, HA show that individual shift values cannot be recovered from the 1D ShiftCrypt Index but in general the right region in the multidimensional manifold of populated shifts is found?

Yes, it does. It also means that a single dimension is not sufficient to recalculate them. Using a 2 (or more) dimensional index would probably allow a more accurate inclusion of these atoms, but the tool would lose the interpretability factor on which our work is based.

File pearsons.txt: Data is repeated multiple times?

We modified the text file

Supplementary Fig. 19: Scale for the y-axis missing? Identical for all panels?

Yes, it is identical for all panels, we omitted it to save some space in the figure

Method_Explanation.pdf: Explain the mathematical relation between the ShiftCrypt Index and the black lines in the 2nd and 3rd Figures. Is the last paragraph the answer to my question above on inverting the direction of the ShiftCrypt Index for certain residue types?

Yes it was. We modified it in order to make it clearer. We also removed the list of residues from the text, since the fact we inverted them is given by chance only and it may lead to misunderstandings.

If so, why does the neural network intrinsically “switch to inverted values” for these residues?

We agree that this point may be misunderstood. The network indeed chooses the direction randomly (based on the initial random initialization and on the optimizer implementation). We just invert the ones in which the helix propensity is higher on the right side. This does not influence the properties of the ShiftCrypt Index, but it makes it easier to understand and to interpret it for the users.

REVIEWERS' COMMENTS:

Reviewer #3 (Remarks to the Author):

The authors have responded to all remarks.
I recommend publication.